# Investigation of the Effect of Copper Addition on Physical and Mechanical Properties of TiNi-Cu Porous Alloy

Maria Kaftaranova [1], Valentina Hodorenko [1], Sergey Anikeev [1,*], Nadezhda Artyukhova [1], Anastasiia V. Shabalina [1,*] and Victor Gunther [2]

[1] Siberian Physical-Technical Institute, National Research Tomsk State University, 634050 Tomsk, Russia
[2] Faculty of Physics, Tomsk State University, 634050 Tomsk, Russia
* Correspondence: anikeev_sergey@mail.ru (S.A.); shabalinaav-1985@yandex.ru (A.V.S.)

**Abstract:** This work is devoted to the physical and mechanical properties of porous alloys based on TiNi alloyed with different amounts of Cu additive. We show that by doping a porous TiNi alloy with copper instead of nickel, it is possible to obtain characteristics acceptable for use in implantology and superior to those of known porous TiNi alloys. Cu addition in the range from 1 to 10 at.% is shown to optimize the properties of tested alloys. There is a decrease in the minimal martensitic transformation stress $\tau_{M_s}^{min}$ from 37 to 17 MPa when compared to initial unalloyed TiNi. Alloys with 3 and 6 at.% of Cu are found to be optimal for use in medical practice. Along with a wide temperature range of reversible deformations that cover the range of operating temperatures (273–313 K), such alloys demonstrate their martensitic transformation stress values below 28 MPs. This permits to model implantable structures of complex configuration from such materials under a certain temperature regime.

**Keywords:** TiNi based alloy; alloying; copper addition; strength; ductility; martensitic transformation stress; porous alloy; phase transformation





## 1. Introduction

Porous alloys based on titanium nickelide (TiNi) obtained by the method of self-propagating high-temperature synthesis (SHS) are bright representatives of the materials that are widely used for implantation [1,2]. Porous-permeable implants based on TiNi were found to meet both medical and technical requirements. They also have optimal physical and mechanical properties, and demonstrate a high similarity of their pore space with the bone tissues of a living organism [3–10]. Moreover, TiNi alloys are known to exhibit superior wear and corrosion resistance [11].

Developing materials with an optimal set of properties requires control of the following main characteristics of porous alloys: the value of deformation before fracture $\varepsilon_{max}$, the value of the fracture stress $\sigma_{max}$, the temperature interval of the shape memory effect and superelasticity, maximal $\tau_{max}^{M_d}$ and minimal $\tau_{min}^{M_s}$ martensitic transformation stresses, the difference between the critical martensitic transformation stresses $\tau_{max}^{M_d} - \tau_{min}^{M_s}$, Young's modulus $E$. As the parameter $\tau_{max}^{M_d} - \tau_{min}^{M_s}$ increases, it becomes less likely to quickly reach the yield strength of the alloy, thereby the martensitic transformation in the material occurs in a greater extent. Since the total cumulative strain is determined by the sum of the elastic, martensitic, and plastic components, the increase in the martensitic strain component contributes to the increase in the total cumulative strain [1].

Modeling of complex-shaped implants at a certain temperature mode and ensuring the proper cosmetic effect when restoring the function of a lost organ is possible from elastic, easily deformable porous plates. This can be realized in alloys with the minimal martensitic transformation stress $\tau_{min}^{M_s}$, which is responsible for the rigidity of the system, having a low value in the operating temperature range of 273–313 K.

A directed change in the characteristics of porous alloys listed above is possible by varying the concentration of titanium and nickel, by conducting thermomechanical treatment, and by alloying [12]. To date, alloying TiNi with various elements was studied. For example, the authors of work [13] established that in the TiNi-Fe alloys (with 0.5–3.0 at.% of Fe) an increase in Fe content led to a decrease in the amount of the $Ti_3Ni_4$ and $Ti_2Ni$ phases. In our previous work [14], the introduction of Co and Ni activating additives (0.5–2.0 at.%) into the sintered TiNi powder system demonstrated the possibility to effectively control the parameters of porosity and average pore size in a wide range of values. The authors of ref. [15] revealed that TiNiMo alloys had high technological characteristics and were easy to process. Moreover, the TiNi alloys with Ag additives showed good biocompatibility, corrosion resistance and antibacterial activity [16–19]. Additionally, the nickel-rich $Ti_{29.4}Ni_{50.6}Hf_2$ alloys were used as high temperature shape-memory alloys [20]. A partial replacement of titanium with zirconium was reported to increase the martensitic transformation temperatures by more than 400 K [21], while doping the original TiNi with tantalum made it possible to block the release of Ni ions into the human body [22]. The addition of Pd to the TiNi alloy resulted in increased cyclic stability of shape memory characteristics [23], whereas adding Ni ($Ti_{46.9}Ni_{50.1}Nb_3$) led to a well-pronounced shape memory effect in the material [24].

Copper is also an effective alloying element [25–28]. It is known that the introduction of Cu additives into TiNi provides the opportunity to control physical and mechanical properties, as well as the temperature interval and the hysteresis of the shape memory effect appearance [29–32]. The phases precipitated in the structure of alloys during alloying can control the processes of nucleation and growth of martensite crystals, strengthen the austenitic matrix TiNi(B2), act as stoppers during the interphase boundary's movement, and also be the preferential locations for the martensite crystals' nucleation [1,32–35]. It is known that at very low copper additions (up to 1 at.%) instead of nickel, the temperature of the end of the martensitic transformation $M_f$ decreases, and the range of the direct martensitic transformation expands [1]. In the work on ternary alloys $Ti_{50}Ni_{50-x}Cu_x$ ($0 \leq x \leq 40$ at.%) [35], it was shown that the mechanical properties decreased with copper addition, namely the tensile strength, and the value of pseudoelastic deformation. At the same time, the relative elongation to failure remained at a high level (>20%). Li et al. [35] showed that copper alloying of monolithic TiNi alloys above 10 at.% led to a decrease in their manufacturability and an increase of brittleness after exceeding the martensitic deformation. Along with this, when TiNi is alloyed with copper up to 10 at.%, an increase in wear resistance of monolithic NiTiCu alloys at temperatures from 37 °C to 250 °C, as well as an increase in corrosion resistance were shown. An increase in corrosion resistance permits long-term implantation of structures made from this material.

Since the introduction of Cu additives showed a positive effect on the properties of monolithic TiNi alloys, the authors of the present work suggest that the alloys of the $TiNi_{50-x}Cu_x$ system obtained by the SHS method will also exhibit practically useful properties. The number of studies on the properties of porous TiNiCu alloys obtained by the SHS method is limited. Additionally, the currently available literature on TiNiCu alloys is aimed at studying the structural characteristics of TiNiCu alloys [36], characteristics of martensitic transformations, shape memory effects and physical and mechanical properties [32,37]. No articles on the effect of copper additives on martensitic transformation stress were found. However, this characteristic is important for the practical use of porous materials, since it determines the stress at which the first martensite crystals appear. Control of martensitic transformation stress in a wide temperature range can allow researchers to create materials for specific purposes in maxillofacial surgery when replacing a defect in the orbit and midface. That is why, we have planned a set of studies aimed at studying this characteristic, and the present work is of high practical importance.

## 2. Materials and Methods

In this work, porous TiNi$_{50-x}$Cu$_x$ alloys (where x = 0, 1, 3, 6, 10 at.%) obtained by the SHS method using titanium (PTOM), nickel (PNK-10T2) and copper (PMS-1) powders (Figure 1) were studied.

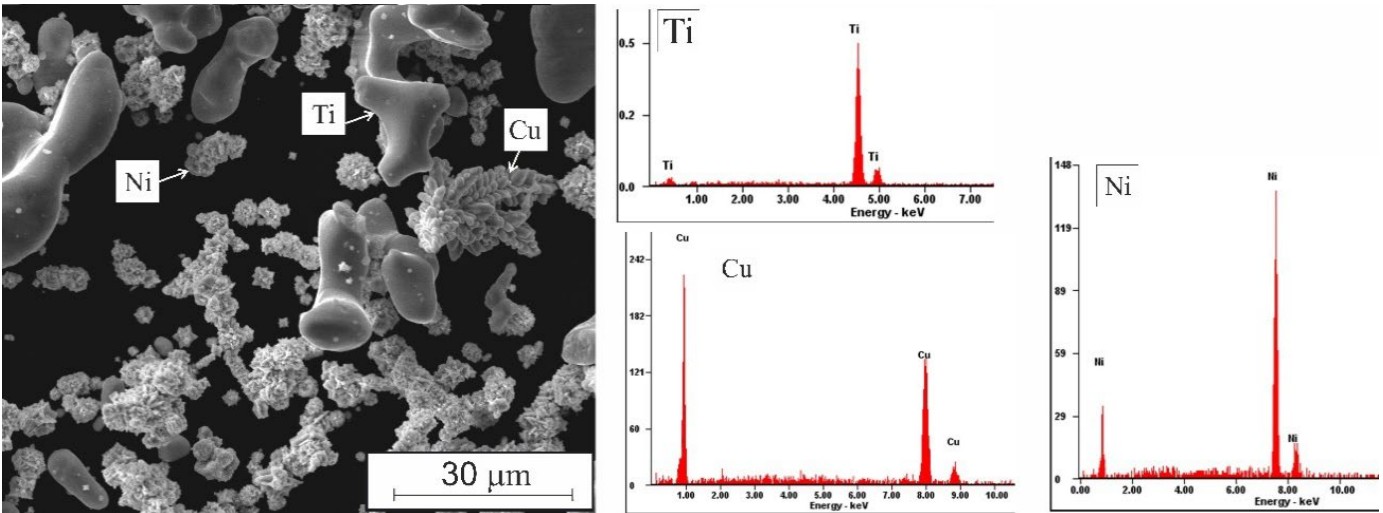

**Figure 1.** Macrostructure of Ti, Ni, and Cu powders with EDX spectra for each component.

The initial Ti, Ni, and Cu powders were dried in a vacuum oven at a temperature of 350–360 K for 7 h; then they were mixed in a mixer for about 8 h. The resulting mixture was poured into a quartz flask, compacted for 30 min by the apparatus shaking method, and then placed in an electric furnace. When the synthesis starting temperature of 743 K was reached, SHS was initiated by a shortcut circuit at the opened end of the Ti-Ni-Cu powdered billet.

Metallographic sections were prepared to study the macro- and microstructure of the alloys. To reveal the microstructural features, the samples were subjected to etching. For this, they were immersed in an acidic solution 3H$_2$O + 2HNO$_3$ + 1HF, then washed with water and alcohol. Metallographic studies were performed using an Axiovert-40MAT optical microscope (Carl Zeiss, Oberkochen, Germany). The microstructure features and chemical composition of the alloys were studied on a SEM 515 scanning electron microscope (Philips, Amsterdam, Netherlands) with using an EDAX ECON IV microanalyzer (EDAX, Mahwah, NJ, USA). The phase composition was studied using X-ray diffraction analysis on a Shimadzu XRD 6000 diffractometer (Shimadzu, Kyoto, Japan).

The sequence and characteristic temperatures of the MT were determined by studying the temperature dependence of the electrical resistance (ER) on a SIES-30 setup. Experimental samples for ER studies of 45 mm × 5 mm × 1 mm were prepared using an electroerosive machine ARTA 153 (SIC Delta-Test, Moscow, Russia). The temperature measurement error was within ±2 K.

Bending strength $\sigma_{max}$ and strain to failure $\varepsilon_{max}$ of porous alloys were determined from the curves of $\sigma(\varepsilon)$ using an Instron 3369 setup (Instron, Massachusetts, UK). The tests were carried out in accordance with the procedures previously described elsewhere [38]. The porous samples were deformed by bending to failure at temperatures of 77, 299, and 423 K.

The main characteristics of shape memory effects: M$_s$ and M$_d$ temperatures, minimal and maximal martensitic transformation stresses $\tau_{max}^{M_d}$ and $\tau_{min}^{M_s}$ were determined using the temperature dependence of martensitic transformation stress $\tau(T)$. To study the temperature dependence of the critical martensitic transformation stresses in a wide temperature range, the samples were deformed by 1.5% at a temperature below the temperatures of MTs and, without unloading, they were heated with simultaneous measuring of the level of

stresses developed by the sample in its attempt to restore its original shape. The obtained information allows qualitative and quantitative assessment of the characteristics of the shape memory effects and superelasticity. The temperature $M_d$ (the maximal temperature at which martensite can be formed under stress) was determined from the maximum on the temperature dependence of the developed forces. Additionally, the temperature $M_s$ was determined from the minimum on this curve. The maximal and minimal martensitic transformation stresses $\tau_{max}^{M_d}$ and $\tau_{min}^{M_s}$ were determined from the stress at the corresponding points $M_d$ and $M_s$. Each point on the $\tau(T)$ curve corresponds to the stress values determined from the $\sigma(\varepsilon)$ curves at different temperatures. The type and nature of the $\tau(T)$ curve provides a complete and qualitative assessment of the behavior of the material in a wide temperature range. Ten samples of 2.5 mm $\times$ 2.5 mm $\times$ 35 mm were used in every experiment.

## 3. Results and Discussion

### 3.1. Macrostructure of the Pore Space

The SHS process is based on the use of heat released as a result of an exothermic reaction between reacting components when an exothermic reaction is initiated in a certain local volume of a substance. Due to thermal conductivity, the released heat heats the layers nearby initiating a reaction in them. The reaction zone moves forward to the heated areas. SHS is possible only with the participation of the liquid phase, which initially forms in the ignition area and then, as a result of an exothermic reaction, propagates along the front of the reaction zone. The reaction zone, in which the reagents are in a liquid state and can be mixed, occupies a thin layer in the charge. The propagation of the reaction and mixing of the melt can occur locally at distances proportionate to the thickness of the reacting layer. After the melt crystallization, interconnected pores are formed in the longitudinal direction of the cylindrical billet.

Figure 2 shows the macrostructure of the $TiNiCu_6$ alloy. This macro-structural image is typical for all porous alloys obtained in this work. The porosity of the materials varies within the range of 65–70%. Analysis of the pore structure of the TiNi alloy showed a characteristic unimodal pore size distribution with the average value being of 100–150 $\mu$m. With an increase in Cu additive from 1 to 10 at.%, the pore size increased from 300 to 500 $\mu$m.

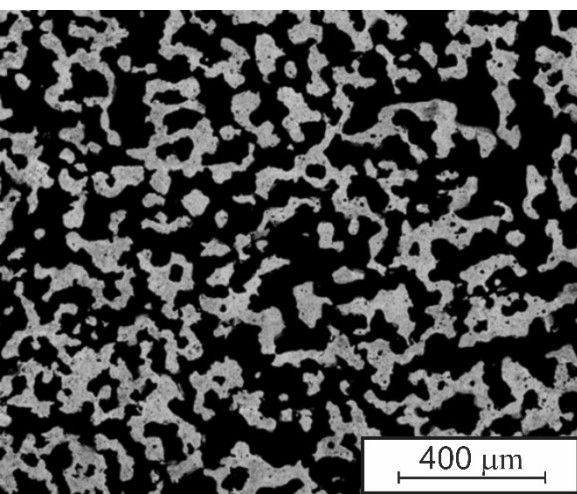

**Figure 2.** Macrostructure of the porous $TiNiCu_6$.

The observed increase in pore size along with the Cu additive concentration is explained by an additional portion of liquid phase formed in the synthesis wave. Its appearance is associated with a number of intermetallic compounds forming in the Ti–Cu system (such as $Ti_3Cu_4$, $Ti_2Cu_3$, and $TiCu_2$) and whose melting temperature lies below

the melting temperatures of the Ti–Ni (Ti$_2$Ni) phases. So, the formation of intermediate phases containing Cu can explain the increase in the volume of the melt during SHS. An increase in the volume of the melt leads to larger interpore bridges in the structure of the solid framework, which is accompanied by an increase in pore size.

### 3.2. Phase Composition of the TiNi Alloys with Various Additions of Cu

According to XRD patterns (Figure 3), the TiNi alloy has two phases of TiNi (B2 and B19′) and Ti$_2$Ni phases (of 0.1–4.3 µm). The porous structure of TiNiCu$_1$–TiNiCu$_{10}$ alloys is typical for all porous materials obtained with the participation of a liquid phase. It was found that, along with the TiNi in the two-phase state B2 and B19′, the alloys also contained phases enriched in titanium (Ti$_2$Ni). Copper completely dissolves in the TiNi matrix phase, which is why its individual particles were not found. Using XRD it was also found that an increase in Cu additive up to 10 at.% leads to an increase of the fraction of the TiNi(B2) phase from 18.3 to 33.9%.

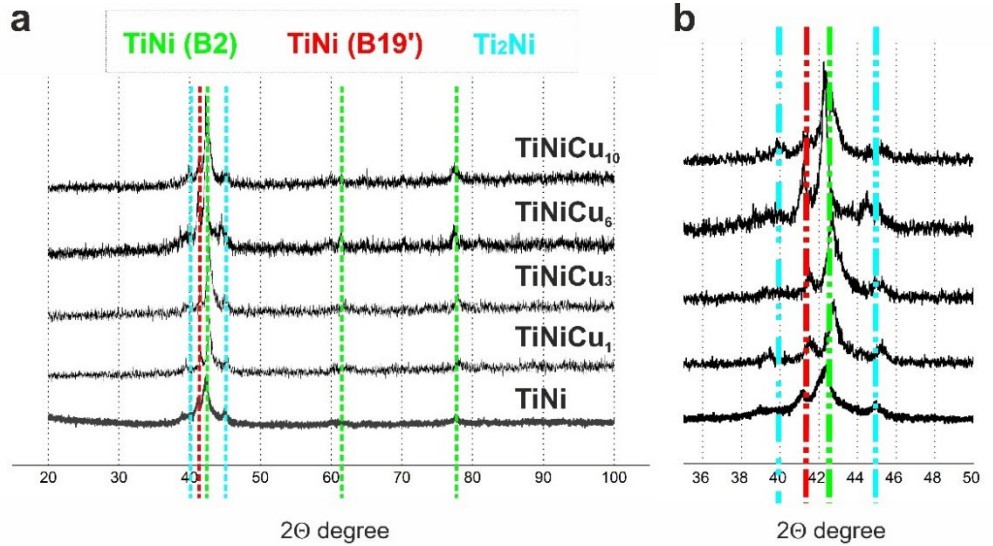

**Figure 3.** X-ray diffraction (XRD) patterns (**a**) and their enlarged range 38–48° (**b**) for the materials.

Doping with Cu led to significant structural changes in the porous TiNi alloy. The formation of dendritic regions occurred, which indicates a pronounced phase-chemical inhomogeneity (see Figure 4 and Table 1). An increase in Cu additive from 1 to 10 at.% also led to a change in the amount of material occupied by dendritic structures. When alloyed with 1–3 at.% of Cu, majority of the matrix bulk was occupied by small, randomly located dendrites. With an increase in Cu content up to 10 at.%, larger dendrites were formed. They did not fill the entire bulk of the material but are seen located in separate parts of the matrix in the center of interpore bridges (Figure 3).

Along with an increase in the size of pores and interpore bridges, an enlargement of the bodies of dendrites and interdendritic layers is observed. As can be seen in Table 1, the difference in the chemical composition of the TiNi compound in the body of the dendrite and the matrix part increases along with Cu addition due to the different ratio of Ni and Cu. At the same time, the concentration of Cu in the matrix remains practically unchanged, being within 1.9–2.5 at.%, while it varies in the body of the dendrite from 1.4 to 7.9 at.%. This feature contributes to the overall level of phase-chemical inhomogeneity of SHS materials based on TiNiCu and can contribute to the expansion of the temperature ranges of their martensitic transformations.

Thus, the study of the structure of porous alloys showed that with the Cu addition increase from 1 to 10 at.%, the individual features of each alloy are determined by the size, distribution and density of precipitated particles and dendrites.

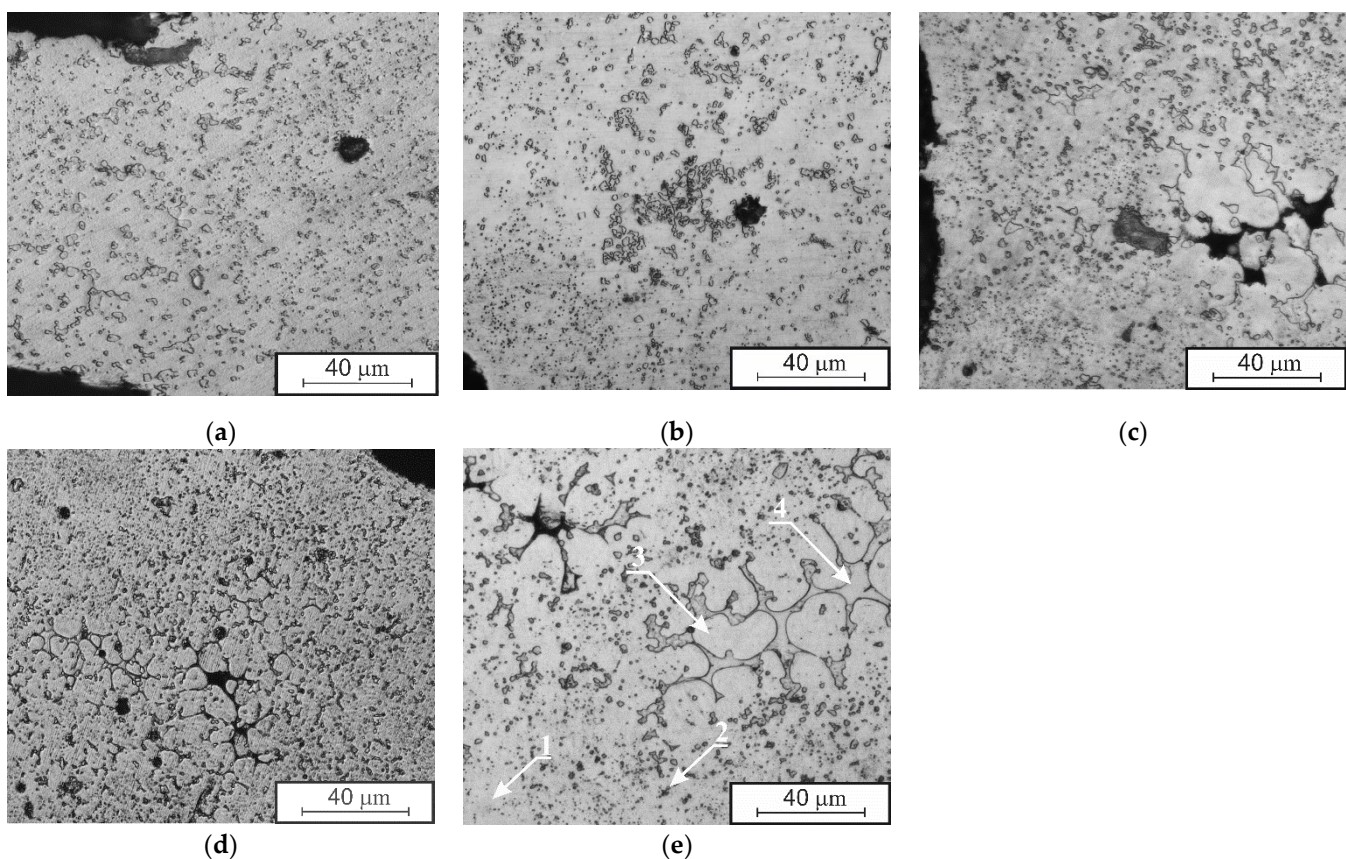

**Figure 4.** Microstructure of porous alloys: (**a**) TiNi; (**b**) TiNiCu$_1$; (**c**) TiNiCu$_3$; (**d**) TiNiCu$_6$ (**e**) TiNiCu$_{10}$: 1—matrix phase TiNi(B2), 2—Ti$_2$Ni, 3—dendrite body, 4—interdendritic layer.

**Table 1.** Chemical composition of porous alloys.

| Element | | Composition in at.% | | |
| --- | --- | --- | --- | --- |
| | | Matrix | Dendrite Body | Interdendritic Layer |
| TiNi | Ti | 49.6 ± 0.9 | - | - |
| | Ni | 51.4 ± 0.9 | - | - |
| | Cu | - | - | - |
| TiNiCu$_1$ | Ti | 49.7 ± 0.6 | 49.9 ± 0.6 | 63.9 ± 0.6 |
| | Ni | 48.4 ± 0.6 | 48.7 ± 0.6 | 34.5 ± 0.6 |
| | Cu | 1.9 ± 0.6 | 1.4 ± 0.6 | 1.6 ± 0.6 |
| TiNiCu$_3$ | Ti | 48.7 ± 0.6 | 49.1 ± 0.6 | 59.9 ± 0.6 |
| | Ni | 49.4 ± 0.6 | 47.6 ± 0.6 | 37.8 ± 0.6 |
| | Cu | 1.9 ± 0.6 | 3.3 ± 0.6 | 2.3 ± 0.6 |
| TiNiCu$_6$ | Ti | 48.5 ± 0.8 | 49.6 ± 0.8 | 63.2 ± 0.8 |
| | Ni | 49.0 ± 0.8 | 45.4 ± 0.8 | 31.3 ± 0.8 |
| | Cu | 2.5 ± 0.8 | 5.0 ± 0.8 | 5.5 ± 0.8 |
| TiNiCu$_{10}$ | Ti | 48.3 ± 0.7 | 49.0 ± 0.7 | 64.6 ± 0.7 |
| | Ni | 49.2 ± 0.7 | 43.1 ± 0.7 | 30.5 ± 0.7 |
| | Cu | 2.5 ± 0.7 | 7.9 ± 0.7 | 4.9 ± 0.7 |

*3.3. Temperature Dependence of Electrical Resistivity R(T) and Critical Martensitic Transformation Stresses τ(T)*

The study of the temperature dependence of electrical resistivity R(T) allowed us to determine and analyze the characteristic temperatures of MTs of porous TiNi alloys with the copper addition (Table 2). It was revealed that with an increase in Cu content from 1 to 10 at.%, an intensive expansion of the temperature range of phase transformations was observed (Figure 5). Using EDX microanalysis it was found that with an increase of Cu

content up to 10 at.%, the Ni content in the dendrite body decreased from 48.7 to 43.1 at.%, while that of Cu changed from 1.4 to 7.9 at.%. On the contrary, the composition of the matrix at 10 at.% of Cu was 48.3 at.% of Ti, 49.2 at.% of Ni, and 2.5 at.% of Cu. Taking into account the increase in the size of dendritic structures in the interpore bridges, there is an increase in the size of areas with chemical composition different from the matrix. As a result, a decrease and broadening in the characteristic temperatures of the MT is observed.

**Table 2.** Characteristic temperatures of MTs for porous alloys.

| Alloy | Temperature, K | | | | |
|---|---|---|---|---|---|
| | $M_s$ | $M_f$ | $A_s$ | $A_f$ | $\Delta T$ |
| TiNi | 338 | 298 | 348 | 388 | 25 |
| TiNiCu$_1$ | 323 | 283 | 333 | 345 | 36 |
| TiNiCu$_3$ | 323 | 263 | 319 | 349 | 41 |
| TiNiCu$_6$ | 313 | 243 | 313 | 353 | 55 |
| TiNiCu$_{10}$ | 283 | 158 | 288 | 353 | 100 |

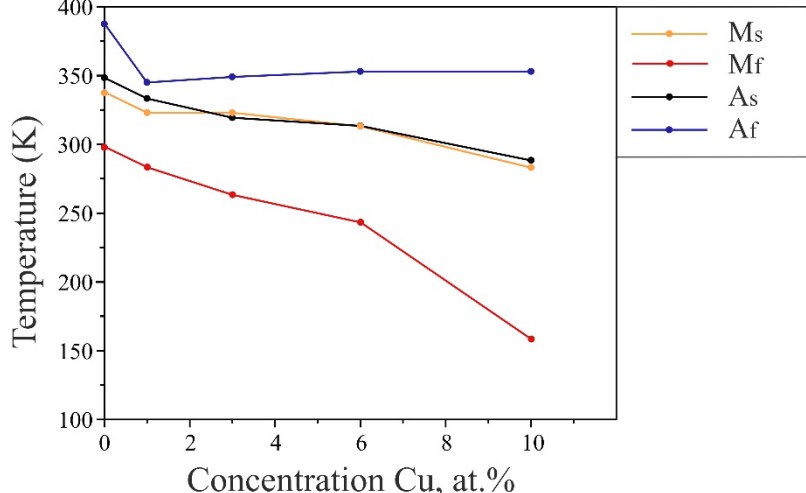

**Figure 5.** Concentration dependence of the characteristic temperatures of martensitic transformations for materials TiNi-Cu.

The use of porous TiNi alloys in medicine involves the modeling of implants with complex configurations in the operating temperatures of 273–313 K. This can be realized in easily deformable porous alloys having minimal system rigidity in a wide temperature range [1,35]. In TiNi alloys, the rigidity of the system is determined by the minimal value of the martensitic transformation stress. The lower the value of $\tau_{min}^{M_s}$, the lower the rigidity of the system. For example, an endoprosthesis that is made of porous TiNi with low martensitic transformation stress. It is quite flexible and can be further adapted more accurately to the defect when replacing the bone structure of the orbit and other midface defects.

Analysis of the temperature dependence of the critical martensitic transformation stresses $\tau(T)$ showed that the porous TiNi alloy is characterized by a wide temperature range of the phase transition appearance. This expansion of the temperature range is associated with the phase-chemical inhomogeneity of porous alloys formed in the SHS process (Figure 6). The MT does not occur in the entire bulk of the material at the same time. This process gradually continues in a certain temperature range. Due to the presence of fine particles, the phase transformations begin in different local areas at different temperatures.

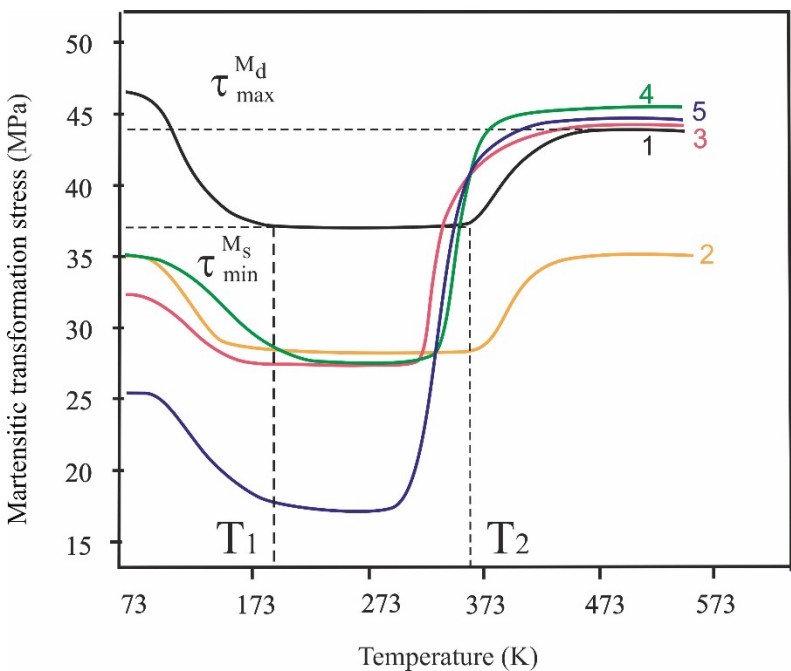

**Figure 6.** Temperature dependence of martensitic transformation stresses: 1—TiNi, 2—TiNiCu$_1$, 3—TiNiCu$_3$, 4—TiNiCu$_6$, 5—TiNiCu$_{10}$.

Figure 6 shows the temperature dependences of the critical martensitic transformation stresses τ(T) of porous TiNi alloys. Along with a wide temperature range of the phase transition covering the range of operating temperatures from 273 to 313 K, porous alloys with 1–6 at.% of Cu are characterized by a martensitic transformation stress below $\tau_{min}^{M_s}$ 30 MPa (Table 3). Alloying porous TiNi with 10 at.% of copper leads to a decrease of $\tau_{min}^{M_s}$ to 17 MPa. However, the range (T$_1$–T$_2$) becomes narrower when compared with alloys TiNiCu$_1$–TiNiCu$_6$.

**Table 3.** Minimal and maximal martensitic transformation stresses of porous TiNi alloys with copper addition.

| Alloy | Temperature, K | | |
|---|---|---|---|
| | $\tau_{M_s}^{min}$, MPa | $\tau_{M_d}^{max}$, MPa | $\tau_{max}^{M_d} - \tau_{min}^{M_s}$, MPa |
| TiNi | 37 | 44 | 7 |
| TiNiCu$_1$ | 29 | 35 | 6 |
| TiNiCu$_3$ | 28 | 45 | 17 |
| TiNiCu$_6$ | 27 | 47 | 20 |
| TiNiCu$_{10}$ | 17 | 45 | 28 |

A high value of $\tau_{min}^{M_s}$ reduces the plasticity of implants, and the possibility of their modeling to the configuration of replaced tissue fragments. The perspective of obtaining porous alloys with a low $\tau_{min}^{M_s}$ in the temperature range of 273–313 K, which is typical for a functioning organism, opens up a wide range of possibilities for their use to replace tissue defects [35].

Along with the $\tau_{min}^{M_s}$ and (T$_1$–T$_2$), an important characteristic is also the difference between the maximal and minimal martensitic transformation stresses $\tau_{max}^{M_d} - \tau_{min}^{M_s}$ [1]. The total deformation of TiNi materials consists of elastic, martensitic and plastic components. The lower the contribution of plastic deformation to the total deformation of the material, the more complete implementation of the shape memory effects and the lower the amount of metal under return to the original form. The reversible accumulation of martensitic deformation, which is the base of the shape memory effects, appears in the temperature range

of MTs below the temperature $M_d$. In this interval, the stressed states of the martensitic and high-temperature phases at a certain temperature coincide. Additionally, they correspond to the martensitic transformation stress, which is determined by the martensitic and elastic deformation. However, the closer to $M_d$ the material is deformed, the more intense the plastic deformation processes are. Near the $M_d$ temperature even a slight deformation of the alloy leads to the stress relaxation through not only the formation of a martensite phase, but also due to the plastic shear. At the $M_d$ temperature, the martensitic transformation stress coincides with the macroscopic yield strength. Above $M_d$ the deformation is performed only through the plastic shear.

Alloying of porous TiNi alloys with copper leads to a decrease in the minimal $\tau_{min}^{M_s}$ and an increase in the maximal $\tau_{max}^{M_d}$ martensitic transformation stresses. As the $\tau_{max}^{M_d} - \tau_{min}^{M_s}$ increases, the main contribution to the deformation of the material is provided by the martensitic deformation (Table 3).

In the $TiNiCu_6$ and $TiNiCu_{10}$ alloys, the minimal martensitic transformation stresses near the $M_s$ temperature and in the range of MT temperatures are low, while the yield strength is high. The appearance of a plastic component of deformation in alloys in this temperature range is unlikely. The contribution of the plastic component to the total deformation can be observed only near the $M_d$ temperature. In TiNi, $TiNiCu_1$, and $TiNiCu_3$ alloys the $\tau_{max}^{M_d} - \tau_{min}^{M_s}$ has minimal values; so, for them the rapid achievement of the yield strength would cause the plastic deformation. This leads to the decrease in the value of the total accumulated deformation of the alloys.

Table 3 shows that increasing the Cu content results in a significant decrease in the value of the $\tau_{M_s}^{min}$. It changes from 37 to 17 MPa when compared with the initial unalloyed TiNi. This is due to the features of the forming alloy structure. At the same time, the value of $\tau_{M_d}^{max}$ remains almost unchanged.

Thus, an important result which has the prospect of further practical applications is the determination of the range of optimal copper concentrations. We found that the addition of 3–6 at.% of copper resulted in a wide temperature range of reversible deformations, along with the low martensitic transformation stress. This finding is of a great practical value, since from the point of view of reconstructive surgery, the decrease in the minimal martensitic transformation stress can allow more accurate modeling of large and complex implants for use in living tissue defects reconstruction.

*3.4. Deformation Dependence σ(ε)*

Analysis of the temperature dependence of electrical resistivity allows determination of the temperatures to study the deformation dependence *σ(ε)*. In this work, the deformation dependences *σ(ε)* were obtained at different temperatures (77, 299, and 423 K). At 77 K the alloys obtained are in the martensitic state; at 299 K they are in a mixed martensite-austenitic (two-phase) state; at 423 K the alloys are only in the austenitic state. This allows us to analyze the properties of alloys in a wide temperature range.

At a temperature of 299 K (two-phase state), the maximal strain to failure was found to be 4–5.5% (Table 4). The martensite arises under loading. It relaxes the peak stresses in the B2 phase through MT and plastic shear, thus increasing the deformation capabilities of the sample [1]. The fracture stress at this temperature varies from 39 to 53 MPa.

**Table 4.** Fracture stress and strain to fracture values for porous TiNi alloys.

| Temperature, K | 77 | | 299 | | 423 | |
|---|---|---|---|---|---|---|
| $\sigma_f$, **MPa**/$\varepsilon_f$,% | $\sigma_f$ | $\varepsilon_f$ | $\sigma_f$ | $\varepsilon_f$ | $\sigma_f$ | $\varepsilon_f$ |
| TiNi | 48 | 5.5 | 53 | 5.5 | 52 | 3.2 |
| $TiNiCu_1$ | 33 | 5.3 | 39 | 5.5 | 42 | 3.1 |
| $TiNiCu_3$ | 58 | 5.3 | 52 | 5 | 65 | 3.7 |
| $TiNiCu_6$ | 45 | 3.5 | 44 | 4.1 | 52 | 2.4 |
| $TiNiCu_{10}$ | 49 | 2.6 | 49 | 4 | 44 | 1.8 |

The deformation of alloys in the martensitic state at 77 K occurs due to the reorientation of the already existing thermal martensite. Along with this, the twinning of the martensite crystals' structure takes place. The main contribution to the total deformation belongs to the martensitic and plastic components. Elastic deformation exhibits a relatively low value. For the alloys at this temperature, the maximal deformation before failure is found to be of 2.6–5.5%.

The deformation at 423 K is 1.8–3.7%. Here, it is determined only by the plastic component of the deformation. In the austenitic state, the deformation mechanism associated with MTs is absolutely excluded. The fracture stress of the alloys at this temperature reaches 42–65 MPa (Table 4).

According to the data obtained using the $\sigma(\varepsilon)$ and $\tau(T)$ curves, the most promising material for the practical use is $TiNiCu_3$. It is characterized with properties acceptable for implantation material, such as the decrease in the minimal martensitic transformation stress responsible for the rigidity of the system, as well as the retention of strength and plasticity values.

## 4. Conclusions

Thus, it was established that the increase in the content of alloying element Cu from 1 to 10 at.% leads to the expansion of the temperature range of phase transformations in TiNi due to the shift of the ending temperature of martensite transactions, $M_f$, to lower values.

It is shown that doping porous TiNi alloy with copper instead of nickel, permits to obtain characteristics acceptable for use in implantology. According to the obtained $\tau(T)$ and $\sigma(\varepsilon)$ curves, the newly prepared alloys with 3 and 6 at.% of Cu were best optimized for use in medical practice. This was due to their wide range of operating temperatures including 273–313 K, and due to decreased values of their martensitic transformation stress below 30 MPa.

It was also established that at a temperature of 299 K, when the alloys are in the two-phase state, the martensite formed under loading relaxes the peak stresses in the B2 phase by means of a martensitic reaction and plastic shear. The deformation of alloys in the martensitic state at 77 K occurs due to the reorientation of the existing thermal martensite. Along with this, the twinning process of the martensite crystals' structure takes place. Deformation at 423 K is determined only by the plastic component. The deformation mechanism associated with MT is completely excluded.

**Author Contributions:** Conceptualization, M.K. and V.H.; methodology, S.A.; software, N.A.; validation, A.V.S., M.K. and V.H.; formal analysis, A.V.S.; investigation, M.K.; resources, V.H.; writing—original draft preparation, M.K.; writing—review and editing, V.H. and V.G.; visualization, A.V.S.; supervision, V.H. and V.G.; project administration, S.A.; funding acquisition, S.A. All authors have read and agreed to the published version of the manuscript.

**Funding:** Investigation of the temperature dependence of electrical resistivity, martensitic transformation stress and strength was funded by Russian Science Foundation, grant number 19-79-10045, https://rscf.ru/project/19-79-10045/ (accessed on 20 August 2022) (contribution of 50%). The study of the features of the microstructure was carried out with the support of the Tomsk State University Development Programme (Priority-2030) (contribution of 50%).

**Data Availability Statement:** Not applicable.

**Acknowledgments:** The analyses (SEM researches) were carried out with the equipment of Tomsk Regional Core Shared Research Facilities Center of National Research Tomsk State University. Center was supported by the Ministry of Science and Higher Education of the Russian Federation Grant no. 075-15-2021-693 (no. 13.RFC.21.0012).

**Conflicts of Interest:** The authors declare no conflict of interest.

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
