# Peer review of "Investigation of the Effect of Copper Addition on Physical and Mechanical Properties of TiNi-Cu Porous Alloy"

_metals, doi:10.3390/met12101696_

Round 1

Reviewer 1 Report

1. Authors must revise the manuscript for grammatical and typo errors.

2. In Table 1, replace the term bal. with 50%, because the amount of Ti is same 50% in all the cases, you are changing only the Ni amount.

3. Line 45 and 307, is the temperature range mentioned as 273¸313 K. What is the sign “¸” stands for. It is confusing.

4. Line 55, in appropriate spacing must be corrected in the reference numbers in the text.

5. In figure 3 XRD, no need to mention the intensity values in the Y-axix. This is just an arbitrary value.

6. Addition of Cu 6%, shows more intense XRD peaks compared to other XRD peaks, what is the reason for that? Is there any maximum crystal growth during the addition of 6% Cu?

Author Response

We like to thank the Reviewers for careful reading of the manuscript and fair remarks, which made it possible to improve the quality of the work.

  1. Authors must revise the manuscript for grammatical and typo errors.

The text was proofread by a fluent English-speaker (fragments highlighted in blue).

  1. In Table 1, replace the term bal. with 50%, because the amount of Ti is same 50% in all the cases, you are changing only the Ni amount.

The table was deleted.

  1. Line 45 and 307, is the temperature range mentioned as 273¸313 K. What is the sign “¸” stands for. It is confusing.

Corrected.

  1. Line 55, in appropriate spacing must be corrected in the reference numbers in the text.

Corrected.

  1. In figure 3 XRD, no need to mention the intensity values in the Y-axix. This is just an arbitrary value.

The figure was replaced.

  1. Addition of Cu 6%, shows more intense XRD peaks compared to other XRD peaks, what is the reason for that? Is there any maximum crystal growth during the addition of 6% Cu?

The increase in the intensity of the XRD peaks may be due to several factors.

– Decreasing the defectiveness of the structure and reducing the level of stresses in the solid TiNi matrix of the porous SHS material.

– The effect of porosity. At this, an increase in the porosity coefficient leads to the less scattering of the X-ray beam, and the secondary X-ray beam retains a greater intensity after interaction with the porous material.

– Increased the size of crystals in the structure of the material and texturing the sample.

– Features of the measuring device.

Experimental porous SHS samples have an inhomogeneous macro- and microstructure. The macro-inhomogeneity of the porous SHS material meaning that the sample consists of a TiNi metal framework and pores. The phase and chemical inhomogeneity, which is characteristic of the SHS technology, makes it even more complicated.

The existing experience with XRD study of the porous SHS materials based on TiNi indicates that the intensity peaks difference from sample to sample can take place. This can be associated both with the internal structure of the material, and (most likely) with the features of the measurement process. Based on the increase in the intensity of the XRD peaks in this case, it is impossible to judge the growth of crystals, and this requires additional deeper study.

Following your comment, we deleted the y-axis from the figure.

Reviewer 2 Report

The authors use the term “martensitic shear stress” which is actually the “martensitic transformation stress”, and has nothing to do with the shear stress. In addition, the symbol they use, σ, represents “axial stress” not “shear stress” which should be represented by τ. I would suggest that “martensitic shear stress” should be corrected as “martensitic transformation stress”

Line 48: Copper is an effective alloying element along with Mo, Fe, Co [12–15].

Authors should elaborate more on this. There are several potential alloying additions to NiTi, e.g. Pd, Hf, Zr, Ag, V, Nb, Ta, etc. Why did the authors specifically select Mo, Fe and Co? Also the given references (12-15) do not show any explicit information about Mo, Fe and Co alloying to NiTi.

Line 60: … the value of pseudoelastic deformation at the site of phase fluidity. At the 60 same time, the relative elongation to destruction remained at a high level (>20%)

What do authors mean by “the value of pseudoelastic deformation” and “phase fluidity? Please be more clear. Also, please use proper terminology, i.e. “fracture” or “failure” instead of “destruction”.

Line 72:” A literature overview has shown that the number of researches aimed at studying 72 the properties of porous TiNiCu alloys obtained by the SHS method is limited”.

Authors should summarize the outcomes of this limited literature.

Line 83: There is no need for Table 1, as it lists the nominal compositions of the selected alloys and gives no new information. The compositions are already described in the first sentence of “2. Materials and Methods”

Line 103: More information is needed about the bending tests. What kind of bending? How were stress and strain values calculated? Strain rate? How were heating and cooling performed? Were the tests performed according to any standard?

Line 140: “With an increase in the dopant additive from 1 to 10 at.% of Cu, the pore size in- 140 creases from 300 to 500 μm, respectively” Can the authors explain why pore size increases with increasing Cu additions?

Figure 3: It is surprising that Ti3Ni4 phases are detected in all of the alloys. Equiatomic NiTiX alloys do not typically contain Ni-rich Ti3Ni4 phases. Could the authors support their XRD results with any SEM or TEM images showing the presence of Ti3Ni4 phases? It is also not very reliable to index XRD peaks as Ti3Ni4, since the intensities are low and those peaks might belong to B19’ martensite or any other precipitates.

Table 2: TiNi element is shown to have a matrix composition of 51.4 at.%. At this composition, you should not see any martensitic transformation in this material but Table 3 shows transformation temperatures above 298K. Could the authors clarify this contradiction?

Table 2: What is the error level of the composition analysis results in Table 2? How many readings were performed for each element?

What is y-axis label in Figure 6? What is the meaning of “strain stress”?

Author Response

We like to thank the Reviewers for careful reading of the manuscript and fair remarks, which made it possible to improve the quality of the work.

The authors use the term “martensitic shear stress” which is actually the “martensitic transformation stress”, and has nothing to do with the shear stress. In addition, the symbol they use, σ, represents “axial stress” not “shear stress” which should be represented by τ. I would suggest that “martensitic shear stress” should be corrected as “martensitic transformation stress”

The text was corrected.

Line 48: Copper is an effective alloying element along with Mo, Fe, Co [12–15].

Authors should elaborate more on this. There are several potential alloying additions to NiTi, e.g. Pd, Hf, Zr, Ag, V, Nb, Ta, etc. Why did the authors specifically select Mo, Fe and Co? Also the given references (12-15) do not show any explicit information about Mo, Fe and Co alloying to NiTi.

The text was corrected, the Introduction was extended.

Line 60: … the value of pseudoelastic deformation at the site of phase fluidity. At the same time, the relative elongation to destruction remained at a high level (>20%)

What do authors mean by “the value of pseudoelastic deformation Please be more clear.

To reply this comment, we would like to cite a work by Parvizi with co-authors [Parvizi S., Hashemi S. M., Moein S., 19 - NiTi shape memory alloys: properties In Micro and Nano Technologies, Nickel-Titanium Smart Hybrid Materials. 2022, 399-426.]:

“Superelasticity is also called “pseudoelasticity” and refers to the condition when the functional temperature is above the austenite finish temperature or is between the Af and As. In this condition, TiNi is in the parent-structure austenite phase. By applying stress, the phase transformation to the sheared derivative structure/stress-induced martensite (SIM) will occur. In this situation the martensite phase is only stable in the presence of stress and by removing the stress, martensite becomes thermodynamically unstable, thus TiNi will reverse to the austenite state and the original shape will be recovered. Therefore, in the case of superelasticity against the SME, no thermal cycling is needed for transformation and large strains applied by loading to SMAs can be recovered by unloading. As can be observed by applying low stress at low temperatures, TiNi will be deformed easily, however at higher temperatures a higher level of stress is required.”

Regarding the used “phase fluidity”, it is more correct to use “yield stress”. However, in the text of the manuscript we deleted this fragment.

Also, please use proper terminology, i.e. “fracture” or “failure” instead of “destruction”.

The text was corrected.

Line 72:” A literature overview has shown that the number of researches aimed at studying 72 the properties of porous TiNiCu alloys obtained by the SHS method is limited”.

Authors should summarize the outcomes of this limited literature.

The text was corrected.

Line 83: There is no need for Table 1, as it lists the nominal compositions of the selected alloys and gives no new information. The compositions are already described in the first sentence of “2. Materials and Methods”.

The table was deleted.

Line 103: More information is needed about the bending tests. What kind of bending? How were stress and strain values calculated? Strain rate? How were heating and cooling performed? Were the tests performed according to any standard?

The tests were carried out in accordance with the procedures described elsewhere [Ashby, M. F Materials selection in mechanical design, 1st ed.; Butterworth-Heinemann:  Kidlington, Oxford, UK, 2011; p. 513]. (This was also added to the text of the manuscript). Please, find more details in the attach pdf-file.

Line 140: “With an increase in the dopant additive from 1 to 10 at.% of Cu, the pore size in- 140 creases from 300 to 500 μm, respectively” Can the authors explain why pore size increases with increasing Cu additions?

We would like to thank the Reviewer for this comment. Indeed, we did not pay enough attention to the explanation of this phenomena. The text was supplemented with the following fragment:

“An increase in the pore size with an increase in the Cu additive concentration is explained by an additional portion of the liquid phase formed in the synthesis wave. Its appearance is associated with the formation of a number of intermetallic compounds in the Ti–Cu system (Ti3Cu4, Ti2Cu3, TiCu2), the melting temperature of which lies below the melting temperatures of the Ti–Ni (Ti2Ni) phases. The formation of intermediate phases containing Cu can explain the increase in the volume of the melt during SHS. An increase in the volume of the melt leads to an enlargement of the interpore bridges in the structure of the solid framework, which is accompanied by an increase in the pore size.”

For additional information, please, find the Ti-Cu binary phase diagram [Massalski T.B.; Okamoto H.; Subramanian P.R.; Kacprzak L. Binary Alloy Phase Diagrams, 2nd Edition, Vol.2, 1st ed.; ASM International, Materials park, OH, USA, 1990; p.1494].

Figure 3: It is surprising that Ti3Ni4 phases are detected in all of the alloys. Equiatomic NiTiX alloys do not typically contain Ni-rich Ti3Ni4 phases. Could the authors support their XRD results with any SEM or TEM images showing the presence of Ti3Ni4 phases? It is also not very reliable to index XRD peaks as Ti3Ni4, since the intensities are low and those peaks might belong to B19’ martensite or any other precipitates.

We are thankful for this comment. Indeed, the intensity of the Ti3Ni4 peaks is low, and it is difficult to reliably judge the presence of these particles. We agreed and removed the Ti3Ni4 from Figure 3.

Table 2: TiNi element is shown to have a matrix composition of 51.4 at.%. At this composition, you should not see any martensitic transformation in this material but Table 3 shows transformation temperatures above 298K. Could the authors clarify this contradiction?

Table 2: What is the error level of the composition analysis results in Table 2? How many readings were performed for each element?

Determination of the dependence of the characteristic martensitic transformation temperatures on the composition of the TiNi compound is a difficult task. Many works are summarized in [Frenzel J., George E.P., Dlouhy A., Somsen Ch., Wagner M.F.-X., Eggeler G. Influence of Ni on martensitic phase transformations in NiTi shape memory alloys. Acta mater. 2010, 58, 3444–3458]. There are the concentration dependences of the temperatures of the onset of martensitic transformation, which were obtained in different periods of research by various authors such as Wang, Hanlon, Wasilewski, Tang in the work by Frenzel et al. So, different authors obtained different Ms temperatures depending on the Ni concentration. A tendency was revealed that with an increase in the nickel concentration, the temperature of the onset of martensitic transformation shifts to the region of lower temperatures (the figure below). In our work, it was found that the TiNi responsible for the martensitic transformation has a composition of 51.4 at.% Ni. The measurement of the chemical composition was carried out for 15-20 times in different parts of the interpore bridges. Therefore, the values of the standard deviation were added to Table 1.

What is y-axis label in Figure 6? What is the meaning of “strain stress”?

Figure 6 was corrected.

By the “strain stress” we meant “martensitic transformation stress” – the stress at which martensite crystals appear under the loading. And Figure 6 shows the temperature dependence of “martensitic transformation stress”.

Reviewer 3 Report

The comments are listed as follows for revision.

1.    Abstract: “Alloys with 3 and 6 at.% Cu are optimal for use in medical practice.” Is this conclusion rigorous? Although the range of optimal Cu concentration (3-6 at.%) was determined by martensitic shear stresses, the fracture stress and strain of TiNiCu6 in Table 5 were poor.

2.    Introduction: “They also have optimal physical and mechanical properties, and demonstrate a high similarity of the pore space with the bone tissues of a living organism” Crucially, NiTi alloy exhibits superior wear and corrosion resistance, and a recent publication can be referred to (Nanomaterials, 2022, 12(4): 705).

3.    Page 2, line 76: “In this work, the porous TiNi50-xCux alloys (where x=0, 1, 3, 6, 10 at.%)….” How to determine the atomic content of Cu (1, 3, 6, 10 at.%)?

4.    Figure 1: The type of powder is not strictly distinguished, please further prove it through EDS analysis. This is also applied to Figure 4.

5.    Page 4, line 139: Interestingly, the pore size of porous alloys changed significantly (300-500 μm) after alloying Cu element with Ti-Ni alloy (100-150 μm). The reasons for these changes need to be explained.

6.    Figure 3: “Using XRD it is also found that an increase in the copper additive up to 10 at.% leads to an increase of the fraction of the TiNi(B2) phase from 18.3 to 33.9%” The quality of Figure 3 is poor, and the change in the fraction of B2 phase cannot be observed.

7.    Page 5, line 160: “Doping with Cu led to significant structural changes in the porous TiNi alloy” With the Cu addition increase from 1 to 10 at.%, the individual features of each alloy are determined by the size, distribution and density of precipitated particles and dendrites. The reason for this statement should be further analyzed.

8.    Figure 5: The significant difference is missed.

Author Response

We like to thank the Reviewers for careful reading of the manuscript and fair remarks, which made it possible to improve the quality of the work.

1.Abstract: “Alloys with 3 and 6 at.% Cu are optimal for use in medical practice.” Is this conclusion rigorous? Although the range of optimal Cu concentration (3-6 at.%) was determined by martensitic shear stresses, the fracture stress and strain of TiNiCu6 in Table 5 were poor.

The task we were faced at was the obtaining a material for reconstructive surgery of the orbit and middle zone of the face. For these purposes, the main requirement for the material is to achieve a “martensitic transformation stress” of less than 30 MPa. The lower this characteristic of the material, the more accurately it is possible to model implantable structures of complex shape from it. The values of the stress of failure and deformation (Table 5, now – Table 4) are sufficient for these purposes.

  1. Introduction: “They also have optimal physical and mechanical properties, and demonstrate a high similarity of the pore space with the bone tissues of a living organism” Crucially, NiTi alloy exhibits superior wear and corrosion resistance, and a recent publication can be referred to (Nanomaterials, 2022, 12(4): 705).

The text was corrected, the reference was added.

3.Page 2, line 76: “In this work, the porous TiNi50-xCux alloys (where x=0, 1, 3, 6, 10 at.%)….” How to determine the atomic content of Cu (1, 3, 6, 10 at.%)?

When preparing the Ti-Ni-Cu powder charge, it was assumed that the introduction of a certain amount of copper in the range of 0–10 at.% corresponds to its content in the sample.

If we get the comment correctly, we provide a formula for calculating the mass of individual components of Ti, Ni, Cu powders to create a mixture for SHS with the specified atomic content of Cu.  Please, find more details in the attach pdf-file.

  1. Figure 1: The type of powder is not strictly distinguished, please further prove it through EDS analysis. This is also applied to Figure 4.

The figure was replaced, EDX spectra were added.

  1. Page 4, line 139: Interestingly, the pore size of porous alloys changed significantly (300-500 μm) after alloying Cu element with Ti-Ni alloy (100-150 μm). The reasons for these changes need to be explained.

The text was supplemented with the following fragment:

“An increase in the pore size with an increase in the Cu additive concentration is explained by an additional portion of the liquid phase formed in the synthesis wave. Its appearance is associated with the formation of a number of intermetallic compounds in the Ti–Cu system (Ti3Cu4, Ti2Cu3, TiCu2), the melting temperature of which lies below the melting temperatures of the Ti–Ni (Ti2Ni) phases. The formation of intermediate phases containing Cu can explain the increase in the volume of the melt during SHS. An increase in the volume of the melt leads to an enlargement of the interpore bridges in the structure of the solid framework, which is accompanied by an increase in the pore size.”

For additional information, please, find the Ti-Cu binary phase diagram [Massalski T.B.; Okamoto H.; Subramanian P.R.; Kacprzak L. Binary Alloy Phase Diagrams, 2nd Edition, Vol.2, 1st ed.; ASM International, Materials park, OH, USA, 1990; p.1494].

  1. Figure 3: “Using XRD it is also found that an increase in the copper additive up to 10 at.% leads to an increase of the fraction of the TiNi(B2) phase from 18.3 to 33.9%” The quality of Figure 3 is poor, and the change in the fraction of B2 phase cannot be observed.

We would like to thank the Reviewer for noticing. The figure was replaced with another one of better quality. It can be seen that the relative intensity of the B2 peaks is growing somehow.

7.Page 5, line 160: “Doping with Cu led to significant structural changes in the porous TiNi alloy” With the Cu addition increase from 1 to 10 at.%, the individual features of each alloy are determined by the size, distribution and density of precipitated particles and dendrites. The reason for this statement should be further analyzed.

The text was supplemented with the following fragment:

“Along with an increase in the size of pores and interpore bridges, an enlargement of the bodies of dendrites and interdendritic layers is observed. As can be seen from Table 1, the difference in the chemical composition of the TiNi compound in the body of the dendrite and the matrix part increases with an increase in the addition of Cu due to the different ratio of Ni and Cu. At the same time, the concentration of Cu in the matrix practically does not change, being at the level of 1.9–2.5 at.%, and varies in the body of the dendrite from 1.4 to 7.9 at. %. This feature contributes to the overall level of phase-chemical inhomogeneity of SHS materials based on TiNiCu and can contribute to the expansion of the temperature ranges of martensitic transformations.”

8.Figure 5: The significant difference is missed.

We are afraid of getting the meaning of the comment incorrectly… According to our understanding of the question, there can be the following reply.

Due to the fact that in the present work the TiNi50-xCux doping formula was used, no significant deviations in the ratio between (Ti) and (Ni+Cu) were observed. Therefore, no sharp difference in temperatures was found in the experimental samples. But due to the increased phase-chemical inhomogeneity, an expansion of the range of martensitic transformations was observed.

Round 2

Reviewer 2 Report

I went through the replies of authors' to my comments and other reviewers. Their revisions are satisfactory for the paper to be published.